# Research on Robust Adaptive RTK Positioning of Low-Cost Smart Terminals

**DOI:** 10.3390/s24051477

**Published:** 2024-02-24

**Authors:** Huizhong Zhu, Jiabao Fan, Jun Li, Bo Li

**Affiliations:** School of Geomatics, Liaoning Technical University (LNTU), Fuxin 123000, China; 472120749@stu.lntu.edu.cn (J.F.); 472010048@stu.lntu.edu.cn (J.L.); libo_5739@lntu.edu.cn (B.L.)

**Keywords:** RTK, GNSS, robust adaptive filter, Kalman filter

## Abstract

The performance of low-cost smart terminals is limited by the performance of their low-cost Global Navigation Satellite System (GNSS) hardware and chips, as well as by the impact of complex urban environments, which affect the positioning accuracy and stability of GNSS services. To this end, this paper proposes a robust adaptive Kalman filter for different environments that can be applied after data preprocessing. Based on the Kalman filter algorithm, a robust estimation approach is introduced into real-time kinematic (RTK) positioning to make judgments on the abnormal observation values of low-cost smart terminals, which amplifies the variance and covariance of the outlier observation equation, and reduces the impact of outliers on positioning performance. The Institute of Geodesy and Geophysics III (IGG III) function is used for regulation purposes, where prior information is modified and refreshed using the equivalent weight matrix and adaptive factors, thus reducing the impact of system model errors on system state estimation results. In addition, a robust factor is defined to adjust positioning deviation weighting between the pre- and post-test robust estimates. The experimental results show that after robust RTK positioning in the static experiments, the overall improvement in positioning accuracies of the Xiaomi 8, Huawei P40, Huawei mate40, and low-cost M8 receiver reached 29.6%, 31.3%, 32.1%, and 30.7%, respectively. Similarly, after applying the proposed robust method in the dynamic experiments, the overall positioning accuracies of the Xiaomi 8, Huawei P40, Huawei mate40, and the low-cost M8 receiver improved by 28.3%, 32.9%, 35.4%, and 26.2%, respectively. The experimental results reveal that an excellent positioning effect of a smartphone is positively correlated with robust RTK positioning performance. However, it is worth noting that when the positioning accuracy reaches a high level, such as the positioning results achieved using low-cost receivers, the robustness performance shows a relatively decreasing trend. This finding suggests that under the condition of high positioning accuracy, the sensitivity of specific positioning equipment to interference sources may increase, resulting in a decline in the effect of robust RTK positioning.

## 1. Introduction

In recent years, GNSS users’ demand for location information has been continuously increasing, and the original users of measurement receivers have been developed into an increasing number of low-cost smart terminal user groups, including smartphones and low-cost receivers. The integration of multiple systems and frequency bands has increased the number of visible satellites and the amount of observation data, thus enhancing the status of smart terminal positioning services. This integration has also improved the data quality, positioning accuracy, and reliability of smart terminals [1]. However, due to the limitations of low-cost chips, the overall data quality of low-cost smart terminals is poor and unstable. Navigation and positioning are often performed in challenging environments, such as urban canyons, where the performance of GNSS positioning is further weakened [2,3]. The increased redundancy of observation data in multiple systems and frequency bands also indicates higher possibilities for gross errors in data [4]. Although observation value quality control can be performed, an actual measurement situation is complex, and it is challenging to control the threshold setting. In high-precision positioning, the carrier phase serves as the main observation quantity, and demands high data quality. However, the quality control method has limitations in its application. If the observation noise is too high, the effectiveness of data verification can be significantly reduced. A robust Kalman filter model offers an improved solution to this problem. It combines robust estimation with the Kalman filter to identify outliers, and reduces their impact on the solution result by increasing the variance and covariance of the outlier observation equation.

In May 2016, Google opened the GNSS raw data acquisition interface for the Android system, enabling domestic scholars to easily access satellite signal measurement values, such as pseudo-range and carrier phase values. To realize accurate location tracking in low-cost smart terminals, various data processing methods and positioning algorithms have been developed [5,6]. Banville [7] studied the GPS single-frequency pseudo-range single-point positioning of low-cost smart terminals. The results showed that low-cost smart terminals could provide positioning data with meter-level accuracy. Gill [8] analyzed the single-frequency precision point positioning (PPP) of low-cost smart terminals, and on the basis of the original single point positioning, the performance analysis of the low cost intelligent terminal antenna has added and proved that low-cost smart terminals could achieve decimeter-level positioning. Chen [9] used a low-cost smart terminal to improve the precision of a single-point positioning mode; the plane positioning accuracy was 0.81 m, and the elevation positioning accuracy was 1.65 m. This research further proves that low-cost terminals can also achieve high-precision positioning. In addition to pseudo-range single-point positioning algorithms, relative positioning algorithms have also been studied; among these algorithms, real-time kinematics (RTK) is the main method [10]. With the increase in the number of systems and frequency bands in low-cost smart terminals, many studies have conducted more in-depth research on their models. Robert [11] used the variance component estimation method to reconstruct a pseudo-range and carrier random model, studied its time correlation, and connected the low-cost smart terminal u-blox M8T to an external professional receiver antenna to achieve the positioning accuracy of a professional receiver. Li [12] considered single-frequency real-time dynamic positioning, added the quartile method to establish a robust model, dynamically determined the threshold, and eliminated the gross error of the observation value. Although abnormal data can be eliminated through data preprocessing and other methods, the quality of GNSS signals received by low-cost smart terminals varies greatly, and it is challenging to control the threshold setting, thus making it impossible to completely eliminate outliers and positioning errors. As a result, the quality control effect in the later period is significantly reduced [13]. Robust adaptive Kalman filtering provides an improved solution, and its essence is to combine robust estimation with Kalman filtering to identify outliers. It increases the variance and covariance of the outlier observation equation to reduce the impact of outliers on the solution [14]. From the aforementioned research, it can be concluded that the quality of observation values of low-cost smart terminals is poor, and abnormal data processing needs to be performed in terms of data preprocessing and quality control to improve positioning performance. In view of that, this research studied robust estimation in smart terminal data processing [15].

In recent years, the Kalman filter has been widely used due to its particularity. Most of the robust Kalman filters have been developed based on the M-estimation approach. Following the Bayesian theory, according to different prior distributions, M-estimation can be divided into three common types: M-LS filtering, LS-M filtering, and MM filtering [16]. Since an observation model is always rank-deficient in the actual settlement process, a robust Kalman filtering algorithm for the rank-deficient model was proposed. In addition, using equivalent weight functions and equivalent quasi-covariance matrix algorithms, the impact of abnormal observations on results can be reduced [17]. Adaptive filtering has also been widely used in recent years, since it can effectively weaken and eliminate the influence of colored noise. The more common adaptive filtering methods include Sage–Husa adaptive filtering and evanescent filtering [18,19]. Sage–Husa adaptive filtering defines a reasonable window, and uses a residual sequence within the window to estimate the observation and system noises in real-time. Evanescent filtering converts the variance–covariance of the previous epoch state vector into the filtering process, and then performs expansion to reduce the impact on the accuracy of parameter estimates. A number of studies have comprehensively considered the systematic errors of observations and kinematic models in the Kalman filtering process, and suppressed the interference of abnormal observation values by adaptively reducing the weight of the equidistant weighted matrix constructed from the observation residuals [20].

Consequently, recent research on robust Kalman filtering has mostly focused on professional receivers, and the theory is relatively mature. However, fewer scholars have used robust adaptive theory to study the positioning problem of low-cost smart terminals. To address this shortcoming, this paper introduces a robust Kalman filter estimation and adaptive algorithm based on the RTK positioning model of a non-difference error correction value. To identify the pre-test abnormal innovation of low-cost smart terminals, the IGGIII function is used for regulation. After undergoing robust processing, the part of the post-test standardized residual that exceeds the threshold is robustly processed. Further, by dynamically adjusting and updating the pre-test innovation of the equivalent weight matrix and adaptive factors, the impact of the system model error on system state estimation is eliminated, thus improving the accuracy of state parameter estimation. In addition, to analyze the improvement effect of the proposed method on low-cost smart terminals, this study uses multi-system multi-frequency data from Xiaomi 8 (Xiaomi, Beijing, China), Huawei P40 (Huawei, Shenzhen, China), and Huawei mate40 (Huawei, Shenzhen, China) smartphones and low-cost receivers. The improvement in the low-cost smart terminal RTK positioning performance achieved by robust adaptive Kalman filtering was analyzed in different environments, positioning states, and system combinations [21]. The results indicate that certain types of smart terminals can achieve high-precision positioning. Finally, feasibility experiments of combined positioning at different frequencies using the same system were conducted for low-cost smart terminals. Therefore, for low-cost intelligent terminal positioning, the RTK method improves the positioning accuracy through differential processing, which can eliminate most of the influence of common error sources of the mobile phone and the receiver. The Kalman filtering algorithm is a filtering method based on state estimation, which is used to estimate state variables of the system. Therefore, the combination of the two methods can provide more accurate location information in mobile phone positioning.

The rest of this paper is organized as follows. Section 2 summarizes the models, chip types, and other characteristics of several low-cost smart terminals that support GNSS raw data, and introduces the multi-system RTK positioning model algorithm and Kalman filter estimation method. Section 3 describes the proposed RTK positioning model based on the Kalman filter’s robust adaptive algorithm, including the construction of adaptive factors. Section 4 verifies the effectiveness of the robust adaptive Kalman filter and the adaptive factor using the static data obtained from several low-cost smart terminals and dynamic data collected under two conditions. Finally, Section 5 concludes this study.

## 2. RTK Positioning Model of Non-Differenced Error Corrections

### 2.1. GNSS Non-Differenced Observations

The non-difference observation equations of the GNSS pseudo-range and carrier phase can be expressed as follows:(1)$$P_{r,j}^{s}=\rho_{r}^{s}+dt_{r}-dt^{s}+I_{r,j}^{s}+T_{r}^{s}+\varepsilon \left(P_{r,j}^{s}\right),$$
(2)$$L_{r,j}^{s}=\rho_{r}^{s}+dt_{r}-dt^{s}+\lambda_{j}\cdot N_{r,j}^{s}-I_{r,j}^{s}+T_{r}^{s}+\varepsilon \left(L_{r,j}^{s}\right),$$
where $P_{r,j}^{s}$ represents the pseudo-range observation value; $L_{r,j}^{s}$ is the carrier phase observation value; r indicates the receiver; j is the frequency of a satellite s; $\rho_{r}^{s}$ is the geometric distance from a receiver r to the satellite s; $dt_{r}$ is the receiver clock error; $dt^{s}$ is the satellite clock error; $I_{r,j}^{s}$ is the ionospheric delay error; $T_{r}^{s}$ is the tropospheric delay error; $\lambda_{j}$ is the wavelength corresponding to frequency j; $N_{r,j}^{s}$ is the integer ambiguity; and ε denotes the observation noise corresponding to the observation value.

It should be noted that errors, such as Earth rotation correction and relativistic effects, are corrected in advance through the error model [22].

### 2.2. Non-Difference Error Correction Value of Base Station

According to the observation Equations (1) and (2), when the coordinates of a reference station are known, the non-difference observation equation of a reference station. r can be expressed as follows:(3)$$E\{{\tilde{P}}_{r,j}^{s}\}=T_{r}^{s}+dt_{r}-dt^{s}+I_{r,j}^{s}+\varepsilon \left(P_{r,j}^{s}\right),$$
(4)$$E\{{\tilde{L}}_{r,j}^{s}\}=T_{r}^{s}+dt_{r}-dt^{s}+\lambda_{j}\cdot N_{r,j}^{s}-I_{r,j}^{s}+\varepsilon \left(L_{r,j}^{s}\right),$$
where ${\tilde{P}}_{r,j}^{s}=P_{r,j}^{s}-\rho_{r}^{s}$ and ${\tilde{L}}_{r,j}^{s}=L_{r,j}^{s}-\rho_{r}^{s}$ represent the observed-minus-computed (OMC) values of the pseudo-range and phase, respectively. 

The OMC value of the non-difference carrier phase is different from that of the pseudo-range. In addition to the integer ambiguity parameters, including error correction information, each error is used to obtain the non-difference error correction value. The meaning of the correction value OSR (observable space representation) is defined by the following equations:(5)$$OSR\{P_{r,j}^{s}\}=E\{{\tilde{P}}_{r,j}^{s}\}=T_{r}^{s}+dt_{r}-dt^{s}+I_{r,j}^{s}+\varepsilon \left(L_{r,j}^{s}\right),$$
(6)$$OSR\{L_{r,j}^{s}\}=E\{{\tilde{L}}_{r,j}^{s}-\lambda_{j}N_{r,j}^{s}\}=T_{r}^{s}+dt_{r}-dt^{s}-I_{r,j}^{s}+\varepsilon \left(L_{r,j}^{s}\right),$$

According to Equation (6), the correction value $OSR\{L_{r,j}^{s}\}$ consists of the receiver clock error $dt_{r}$, satellite clock error $dt^{s}$, ionospheric delay error $I_{r,j}^{s}$, tropospheric delay error $T_{r}^{s}$, and observation noise $\varepsilon \left(L_{r,j}^{s}\right)$. Equation (6) also contains the unknown ambiguity parameter $N_{r,j}^{s}$, which can be absorbed by the user end as part of the error correction value. Since this parameter is an integer, the value of the integer ambiguity does not affect the positioning result of a user station. The same conclusion can be obtained for the error correction value of the pseudo-range observations [23].

### 2.3. RTK Positioning Model of User Station

The non-difference carrier observation equation of s satellites on a user station *u* can be expressed as follows:(7)$$L_{u,j}^{s}=\rho_{u}^{s}+H_{u}^{s}\cdot \delta X+dt_{u}-dt^{s}-\lambda_{j}\cdot N_{u,j}^{s}-I_{u,j}^{s}+T_{u}^{s}+\varepsilon \left(L_{r,j}^{s}\right),$$
where $H_{u}^{s}$ is the coefficient matrix corresponding to the user station position correction value parameter δX; the meanings of other symbols are the same as in Equation (2). 

The observation equation of a user station corrected by the error correction number $OSR\{L_{r,j}^{s}\}$ on a reference station R is defined as follows:(8)$$\begin{array}{l} L_{u,j}^{s}-OSR\{L_{r,j}^{s}\}=\rho_{u}^{s}+H_{u}^{s}\cdot \delta X+\left(dt_{u}-dt_{r}\right)\\ \;\;\;\;\;\;\;\;\;\;-\lambda_{j}\cdot \left(N_{u,j}^{s}-N_{r,j}^{s}\right)-(I_{u,j}^{s}-I_{r,j}^{s})+(T_{u}^{s}-T_{R}^{q}) \end{array},$$

According to Equation (8), the error correction value of a satellite q on a user station u passing through a reference station s can also be obtained after the value of $OSR\{L_{r,j}^{s}\}$ can be obtained, as follows:(9)$$\begin{array}{l} L_{u,j}^{q}-OSR\{L_{r,j}^{q}\}=\rho_{u}^{q}+H_{u}^{q}\cdot \delta X+\left(dt_{u}-dt_{r}\right)\\ \;\;\;\;\;\;\;\;\;\;\;-\lambda_{j}\cdot \left(N_{u,j}^{q}-N_{r,j}^{q}\right)-(I_{u,j}^{q}-I_{r,j}^{q})+(T_{u}^{q}-T_{r}^{q}) \end{array}.$$

Equations (8) and (9) eliminate the satellite clock error. When the baseline between a reference station r and a user station u is short, the atmospheric errors have a good spatial correlation. Equation (9) can also weaken the ionospheric and tropospheric delay errors. After making the difference between the two formulas, Formula (10) can be obtained as follows:(10)$$L_{u,j}^{sq}-OSR\{L_{r,j}^{sq}\}=\rho_{u}^{sq}+H_{u}^{sq}\cdot \delta X-\lambda_{j}\cdot N_{ur,j}^{sq}-I_{ur,j}^{sq}+T_{ur}^{sq}.$$

In Equation (10), the satellite clock error and receiver clock error are eliminated, the ionospheric and tropospheric delay errors are further weakened, and the integer ambiguity is a double difference form with integer characteristics. By selecting different non-differenced reference ambiguities, the obtained rover non-differenced error correction values are completely different. This is because the difference in non-differential ambiguity is a non-differential error correction value. The non-differenced reference ambiguity difference can be transmitted to the rover through the non-differenced error correction value, and then reflected in the rover’s integer ambiguity. Selecting different non-differenced reference ambiguities has no effect on the correction of the rover’s non-differenced errors. Based on (10), the double-difference observation equation for solving the carrier phase integer ambiguity of reference stations A and B can be derived as follows:(11)$$\Delta \nabla L_{AB,j}^{sq}=\Delta \nabla \rho_{AB}^{sq}-\lambda_{j}\Delta \nabla N_{AB,j}^{sq}+\Delta \nabla d_{ion}+\Delta \nabla d_{trop},$$
where Δ∇ represents the double difference operator; and $\Delta \nabla d_{ion}$ and $\Delta \nabla d_{trop}$ are the remaining ionospheric and tropospheric delay residues that have not been eliminated, respectively. 

All experiments presented below denote short-baseline experiments, and the ionospheric and tropospheric delays have a strong correlation, so they can be ignored.

### 2.4. Multi-System RTK Positioning Model

Based on Equation (11), the carrier phase equation of the BDS-3 and GPS systems is expressed as follows [24]:(12)$$\Delta \nabla L_{j}^{B}=\Delta \nabla \rho^{B}-\lambda_{j}\Delta \nabla N_{j}^{B},$$
(13)$$\Delta \nabla L_{j}^{G}=\Delta \nabla \rho^{G}-\lambda_{j}\Delta \nabla N_{j}^{G}.$$

The GLONASS system uses frequency division to distinguish satellites. The signals emitted by different satellites have different frequencies, so the carrier phase observation equation requires mathematical transformation. Assume that P and Q are two satellites observed by the GLONASS system; then, the carrier phase observation equation of the GLONASS system at a frequency j can be obtained as follows:(14)$$\Delta \nabla L_{j}^{P}-\Delta \nabla L_{j}^{Q}=\Delta \nabla \rho -\lambda_{j}^{P}(\Delta N_{j}^{P}-\Delta N_{j}^{Q})-(\lambda_{j}^{P}-\lambda_{j}^{Q})\Delta N_{j}^{Q}.$$

A change in the double-difference ambiguity causes the double-difference ambiguity of the GLONASS system to produce whole-circuit characteristics. This problem can be solved using the positioning mode of the BDS-3 and GPS systems. Then, the normal equation of multiple systems is defined by the following:(15)$$V_{k}=H_{k}X_{k}-L_{k}=\left[\begin{array}{cccc} A^{G} & B^{G} & 0 & 0\\ A^{C} & 0 & B^{C} & 0\\ A^{R} & 0 & 0 & B^{R} \end{array}\right]\left[\begin{array}{c} dX\\ \Delta \nabla N^{G}\\ \Delta \nabla N^{C}\\ \Delta \nabla N^{R} \end{array}\right]-\left[\begin{array}{c} \Delta \nabla L^{G}\\ \Delta \nabla L^{C}\\ \Delta \nabla L^{R} \end{array}\right],$$
(16)$$A=\left[\begin{array}{ccc} \frac{x^{1}-x_{0}}{\rho_{0}} & \frac{y^{1}-y_{0}}{\rho_{0}} & \frac{z^{1}-z_{0}}{\rho_{0}}\\ \vdots & \vdots & \vdots \\ \frac{x^{n}-x_{0}}{\rho_{0}} & \frac{y^{n}-y_{0}}{\rho_{0}} & \frac{z^{n}-z_{0}}{\rho_{0}} \end{array}\right],B=\left[\begin{array}{cccc} \lambda & & & \\ & \lambda & & \\ & & \lambda & \\ & & & \lambda \end{array}\right],B^{R}=\left[\begin{array}{cccc} \lambda^{P} & & & \\ & \lambda^{P} & & \\ & & \lambda^{P} & \\ & & & \lambda^{P} \end{array}\right].$$

The innovation residual $V_{k}$ is obtained from the prediction state vector at a time $t_{k}$. In the above equations, dX represents the coordinate correction vector; G, C, and R are three satellite systems; (x,y,z) denote satellite coordinate data; λ represents the carrier length of the BDS-3 and GPS systems; $\lambda^{P}$ is the carrier length of the GLONASS system; and ρ is the inter-station satellite error geometric distance [25].

## 3. Robust Filtering Algorithm

### 3.1. Robust Kalman Filter Estimation

Pre-test robust estimation refers to performing certain processing on raw data before parameter estimation, in order to reduce the impact of outliers on the estimation result. The innovation vector in Kalman filtering refers to the error sequence between the observed and predicted values. The innovation is the pre-test estimated detection volume. The innovation vector can be expressed as follows:(17)$$X_{k,k-1}=A_{k,k-1}X_{k-1}+W_{k},$$
(18)$$V_{k,k-1}=H_{k}X_{k,k-1}-L_{k},$$
(19)$$Q_{V_{k,k-1}}=H_{k}Q_{X_{k,k-1}}H_{k}^{T}+R_{k},$$
where $X_{k,k-1}$ is the state prediction vector; $V_{k,k-1}$ is the innovation residual vector, which can reflect the error size of an observation; and $Q_{V_{k,k-1}}$ represents the variance–covariance matrix of the innovation vector. In this research, the state noise was assumed to be Gaussian white noise, so the influence of state noise on the algorithm was not considered [26].

The post-test robust estimation refers to modifying the estimation result after parameter estimation to reduce the impact of outliers on the estimation result. The post-test residuals and variance–covariance of the Kalman filter can be respectively expressed by the following:(20)$$V_{k}=R_{k}Q_{V_{k,k-1}}^{-1}V_{k,k-1},$$
(21)$$Q_{V_{k}}=R_{k}Q_{V_{k,k-1}}^{-1}R_{k}^{T}.$$

The post-test standardized residual ${\bar{v}}_{k}(j)$ of the jth residual is to change the diagonal elements in the pre-test variance–covariance matrix $Q_{V_{k,k-1}}$ to the diagonal elements in $Q_{V_{k}}$. However, it should be noted that the standardization method is applicable only if the observations are not correlated with each other, and the variance is distributed diagonally. If the variance matrix is not a diagonal matrix, the covariance elements associated with it need to be adjusted after increasing and inflating the variance of the abnormal observations. It is necessary to ensure that the variance and covariance after inflation can still maintain the original correlation coefficient. 

When performing residual vector testing, the innovation residual vectors need to be standardized. The standardized residuals should conform to the standard normal distribution, and the standardized formula of the j th residual is defined as follows:(22)$${\bar{v}}_{k,k-1}(j)=\frac{v_{k,k-1}(j)}{q_{k,k-1}(j)},$$
where $q_{k,k-1}$ is the square root value of the jth element on the $Q_{V_{k,k-1}}$ diagonal. 

The Huber function or the IGGIII function has been commonly used to determine the excess of the standardized residual and adjust the variance–covariance matrix. In practical applications, the IGGIII function has been more practical than the Huber function. Therefore, this study uses the IGGIII function to control the variance of the observation equation, which can be expressed as follows:(23)$${\bar{R}}_{k}=\left\{\begin{array}{cc} R_{k} & \left|{\bar{v}}_{k,k-1}(j)\right|\leq k_{0}\\ R_{k}\frac{\left|{\bar{v}}_{k,k-1}(j)\right|}{k_{0}}{\left(\frac{k_{1}-k_{0}}{k_{1}-\left|{\bar{v}}_{k,k-1}(j)\right|}\right)}^{2} & k_{0}<\left|{\bar{v}}_{k,k-1}(j)\right|\leq k_{1}\\ 100,000R_{k} & \left|{\bar{v}}_{k,k-1}(j)\right|>k_{1} \end{array}\right.,$$
where *k*_1_ and *k*_0_ are threshold constants. 

Since the residuals in pre-test robust estimation are typically large, based on empirical values, this study set them as follows: *k*_1_ = 7 and *k*_0_ = 2.5. Similarly, in the post-test robust estimation, based on empirical values, they were set as follows: *k*_1_ = 3, *k*_0_ = 1.5.

The equation formed after the inter-star difference is substituted into the observation equation in the RTK positioning contains the reference star observation value. When the variance matrix is not a diagonal matrix, the correlation between elements is inherent in the pre-test. After the variance of abnormal observations is increased and inflated, it is necessary to consider the adjustment of the covariance elements so that the original correlation coefficients of the inflated variance and covariance remain unchanged. Assume that the variances of observation vectors L(i) and L(j) are $\sigma^{2}(i)$ and $\sigma^{2}(j)$, respectively, and the covariance is σ(ij). If gross errors occur in both observation vectors, their variance expansion factors are denoted by λ(i) and λ(j), respectively. Then, the variance after inflation is expressed by the following:(24)$${\hat{\sigma }}^{2}(i)=\lambda (i)\sigma^{2}(i),$$
(25)$${\hat{\sigma }}^{2}(j)=\lambda (j)\sigma^{2}(j).$$

Further, to ensure that the correlation coefficient ρ(ij) is unchanged, the following holds:(26)$$\rho (ij)=\frac{\sigma (ij)}{\sigma (i)\sigma (j)}=\frac{\hat{\sigma }(ij)}{\hat{\sigma }(i)\hat{\sigma }(j)},$$
(27)$$\hat{\sigma }(ij)=\sqrt{\lambda (i)\lambda (j)}\sigma (ij),$$
where $\sigma^{2}(i)$ and $\sigma^{2}(j)$ are the variances after inflation of L(i) and L(j), respectively, and $\hat{\sigma }(ij)$ is the covariance after inflation.

After the Kalman filter provides the estimation result, unlike the pre-test residuals, the post-test residuals show a certain correlation with each other, causing the observation value errors that are not detected by pre-test detection to interfere with the normal value; also, the degree of interference is related to relevance. The post-test robustness may reduce the weight of normal observations and affect positioning performance. To this end, this research adopted the following processing strategy. All gross errors in the pre-test residuals are subjected to robust processing, and normal estimation is performed. During the robust post-test processing, only the maximum value of the standardized residual that exceeds the threshold is subjected to robust processing. If there is a gross error, it is initialized, and the measurement is updated until the standardized residuals are smaller than the threshold.

### 3.2. Robust Adaptive Kalman Filtering

During the Kalman filtering process, there can be large errors in a dynamic model. In this case, the system noise is too small and cannot accurately describe the prediction parameter errors, which can affect the subsequent measurement result of the solution. Even if the robust observation value algorithm is used, it may not be possible to ensure the stability of positioning results. Therefore, to achieve reasonable system noise settings, this study uses an adaptive filtering algorithm to adjust the weight matrix of the state advance prediction vector and observation vector dynamically by constructing an adaptive factor. In addition, the difference between the dynamic model forecast information and the dynamic carrier motion trajectory is eliminated to improve positioning reliability. Moreover, this study performs multi-factor adaptive filtering to regulate the variance of the predicted state quantity. Finally, this study uses multi-factor adaptive filtering to regulate the variance of the predicted state quantity.

The adaptive factor is defined based on the post-test innovation residual. The innovation residual and the components of the predicted state vector discrepancy are expressed as follows:(28)$$V_{k,k-1}(i)=H_{k}{\hat{X}}_{k,k-1}(i)-L_{k},$$
(29)$${\tilde{X}}_{k}(i)={\hat{X}}_{k}(i)-{\hat{X}}_{k,k-1}(i),$$
where the innovation vector $V_{k,k-1}$ is calculated from the predicted state ${\hat{X}}_{k,k-1}$ at a time $t_{k}$; ${\hat{X}}_{k}$ is the state estimate at a time $t_{k}$; $V_{k}$ is the state that has been corrected by $L_{k}$; $V_{k,k-1}$ is the state that has not been corrected by $L_{k}$; therefore, the innovation vector $V_{k,k-1}$ can better reflect the disturbance of the dynamic system; finally, i represents a certain component. 

Then, the calculation formula of adaptive factor $\alpha_{k}$ can be expressed as follows [27]:(30)$$\alpha_{{\hat{X}}_{k}}(i)=\left\{\begin{array}{cc} 1 & \left|{\tilde{V}}_{k,k-1}(i)\right|\leq c_{0}\\ \frac{c_{0}}{\left|{\tilde{V}}_{k,k-1}(i)\right|}{\left(\frac{c_{1}-\left|{\tilde{V}}_{k,k-1}(i)\right|}{c_{1}-c_{0}}\right)}^{2} & c_{0}<\left|{\tilde{V}}_{k,k-1}(i)\right|\leq c_{1}\\ 0 & \left|{\tilde{V}}_{k,k-1}(i)\right|>c_{1} \end{array}\right.,$$
where ${\tilde{V}}_{k,k-1}(i)$ is the standardized residual, and $c_{0}$ and $c_{1}$ are constants. In this study, $c_{0}$ = 1.0 and $c_{1}$ = 2.5; since the prior covariance matrix is a non-diagonal matrix after recursion, the principle of a two-factor equivalent covariance was adopted to ensure that its symmetry and correlation do not change; $\sigma_{{\hat{X}}_{k,k-1}}$ is the standard deviation of a prediction vector ${\hat{X}}_{k,k-1}$. The adaptive covariance matrix of ${\hat{X}}_{k,k-1}(i)$ and the off-diagonal elements of ${\hat{X}}_{k,k-1}(j)$ are related as follows:(31)$${\hat{\sigma }}_{{\hat{X}}_{k,k-1}}(ij)=\sqrt{\alpha_{{\hat{X}}_{k}}(i)}\cdot \sqrt{\alpha_{{\hat{X}}_{k}}(j)}\cdot \sigma_{{\hat{X}}_{k,k-1}}(ij),$$
where the variance of the diagonal element is obtained by the following:(32)$${{\hat{\sigma }}^{2}}_{{\hat{X}}_{k,k-1}}(j)=\alpha_{{\hat{x}}_{k}}(j)\cdot {\sigma^{2}}_{{\hat{X}}_{k,k-1}}(j).$$

Therefore, the overall estimation obtained by the robust adaptive Kalman filter can be expressed by the following:(33)$$P_{k}=(I-K_{k}H_{k}){\bar{P}}_{k,k-1}(I-K_{k}H_{k}{)}^{T}+K_{k}{\bar{R}}_{k}K_{k}^{T},$$
where $P_{k}$ is the process noise, ${\bar{R}}_{k}$ is the equivalent variance, and ${\bar{P}}_{k,k-1}$ is the equivalent variance–covariance matrix. 

The specific process is to adjust the filter gain matrix by changing the respective variance–covariance matrix. After the variance of abnormal observations is increased and inflated, the associated covariance elements L(j) are adjusted so that the original correlation coefficients of the inflated variance and covariance remain unchanged. Further, each standardized residual is adjusted using the IGGIII function to adjust the variance of the observation, in order to achieve robust processing of the post-test residual data.

The flowchart of the robust adaptive filtering process is presented in Figure 1, where $\Delta V_{k}$ is the test information obtained from the innovation vector $V_{k}$ and the covariance matrix. The discrepancy value $\Delta X_{k}$ of the predicted state vector is obtained through the expansion of the state prediction covariance matrix; $\varepsilon_{X}$ and $\varepsilon_{L}$ represent the system threshold and measurement threshold, respectively, which are determined by performing multiple static tests at different points using the motion state equations.

The residual sequence in the observation data is judged based on the observation threshold, and the weight of the observation information is reduced for the part exceeding the threshold. In addition, after the residual sequence in the carrier state data is judged based on the state threshold, adaptive factor correction is performed on the part exceeding the threshold. The process described above constitutes a correction mechanism for the robust adaptive Kalman filter model. In this model, if the observation data residual and the state data residual do not exceed the corresponding thresholds, standard Kalman filtering is performed. This strategy helps to improve the robustness and adaptability of the filtering system so that it can effectively deal with potential abnormal observations and state information.

## 4. Experimental Results and Analysis

### 4.1. Data Processing Policies

This study introduces the speed-constrained multi-frequency robust RTK adaptive Kalman filter algorithm, which performs data preprocessing on the original observation data of low-cost smart terminals. After removing the gross errors using the preset threshold, pseudo-range single-point positioning and Doppler speed measurement positioning are performed to obtain the optimal speed, which is then used in the Kalman filter to update the time. Then, the phase observation data obtained from smartphones and professional receivers are measured and updated, and RTK fixed solutions and floating-point solutions are obtained. In the fixed part of ambiguities, the LAMBDA method is used for ambiguity testing [28]. During the experiment, the residual data beyond the threshold are robustly processed. If there are gross errors, the measurement is initialized and updated again until the standardized residuals are within the threshold.

The smartphones used in this study were Xiaomi 8, Huawei P40, and Huawei Mate40, and the low-cost receiver used was the M8 model (Sinan K803 board) (ComNav Technology Ltd., Shanghai, China). The positioning results obtained in the short-baseline RTK fixed solution state of a professional receiver were used as reference values to analyze the positioning results. The GNSS observation data of multiple systems and frequencies were collected. The device codes, release years, and chip types of the smartphones and low-cost receivers used are shown in Table 1.

The satellite systems and frequency bands that the four low-cost smart terminals could receive are presented in Table 2, where it can be seen that among the smartphones, the Huawei mate40 had the most receivable satellite frequency bands, which was mainly reflected in the received BDS satellite signals. The satellite signals that the low-cost M8 receiver could receive were mostly single-frequency signals [29].

In the experiment (Figure 2), the average number of observable satellites of the Xiaomi 8 multi-system was approximately 25, the average number of observable satellites of the Huawei P40 was 32, and the average number of observable satellites of the Huawei mate40 was slightly higher than that of the Huawei P40 and equaled 35. Among the devices used in the experiment, the Xiaomi 8 had the smallest number of observable satellites, and the low-cost M8 receiver had the largest number of observable satellites. Moreover, the low-cost M8 receiver could capture the strongest satellite signals due to its built-in antenna, but its stability was weak. The satellite signals captured by the smartphones were more stable, and the rate of change in the number of satellites was smaller.

### 4.2. Multi-System Multi-Frequency Differential Static RTK Experiments

The experimental environments were an open balcony environment on a school roof and an open road environment. About 10,000 epochs of GNSS observations were conducted, with a sampling interval of 1 s. The measurement time was 12:38–16:00 on 4 May 2023, and the low-cost smart terminals were connected to each other through a mobile phone holder. The software platform used in this research relies on the improved RTKLIB(b34b) open source code. This study considered the time when the positioning result deviation converged and stabilized within 1 m to reach 60 s as the completion time of convergence [30]. The two data processing schemes used in the experiment were static RTK (scheme 1) and robust static RTK (scheme 2), as shown in Figure 3. All of the experiments in this article were conducted in an open environment.

Figure 3 shows the static RTK solution results of the Xiaomi 8 smartphone. The GPS, BDS, and Galileo systems were used to perform data calculations for multi-system multi-frequency combinations. Through observation, it could be found that the positioning effect of the multi-system, multi-frequency static RTK solution was biased. After robust processing, the accuracy of the static RTK positioning result of the Xiaomi 8 smartphone was improved, and the improvement effect was the most obvious between 4000 epochs and 6000 epochs. The average positioning accuracy in each direction was improved by approximately 1.5 m. The U-direction positioning deviation after around 8500 epochs was also within the range of 0.8 m.

Table 3 shows the positioning deviations of the Xiao Mi 8 smartphone in static RTK positioning and robust static RTK positioning in the multi-system multi-frequency experiments. The results showed that the positioning deviations of static RTK positioning in the E, N, and U directions were 0.857 m, 0.786 m, and 1.068 m, respectively. The deviations of robust RTK positioning in the E, N, and U directions were 0.584 m, 0.523 m, and 0.617 m, respectively. Further, robust RTK positioning accuracies improved by 31.8%, 33.4%, and 32.8% in the E, N, and U directions, respectively; also, the convergence time was reduced by 28 s. Among them, the improvement effect was the best in the N direction.

The static RTK positioning error diagram of the Huawei P40 smartphone obtained under open conditions is presented in Figure 4. Since the Huawei P40 and Huawei mate40 could receive the BDS dual-frequency signals, the BDS dual-frequency was used for observation in the multi-system multi-frequency RTK experiments. The analysis of the Huawei P40 multi-system multi-frequency experimental results showed that the multi-frequency positioning results had converged divergence, and were less volatile after robust processing. Particularly, in epochs 5000–6200, the positioning accuracy improvement was the greatest in all directions, and the positioning deviation in the U direction was maintained at about 0.8 m.

The static RTK positioning results of the Huawei P40 smartphone are presented in Table 4, where it can be seen that the static RTK positioning deviations in E, N, and U directions were 1.104 m, 1.018 m, and 1.530 m, respectively. The deviations of the robust static RTK positioning in the E, N, and U directions were 0.757 m, 0.697 m, and 1.056 m, respectively. Further, compared to the static RTK positioning, the robust static RTK improved the positioning accuracies by 31.4%, 31.5%, and 30.9% in the E, N, and U directions, respectively. Also, the convergence time of the static RTK positioning was 112 s, and the convergence time of the robust static RTK positioning was 87 s, showing an improvement of 25 s.

The static RTK positioning error diagram of the Huawei mate40 smartphone is presented in Figure 5. The results indicate that after robust processing, the positioning deviation was reduced, and the convergence effect was obvious. The convergence time of the Huawei mate40 smartphone in static RTK positioning was 86 s, and that in robust static RTK positioning was 63 s, showing an improvement of 23 s. After 6300 epochs, the static RTK positioning results of the Huawei mate40 smartphone converged, and then diverged. Considering the instability of the smartphone signal, the positioning accuracy was more susceptible to errors, such as atmospheric errors, compared to professional receivers. Around 4000–6000 epochs, the robust improvement effect was the best, and the positioning deviation was maintained at approximately 0.3 m on average. In epochs 4000–6000, the robust improvement effect was the best, and the positioning deviation was maintained at approximately 0.3 m on average.

The deviations of static RTK and robust static RTK positioning are presented in Table 5. The results showed that the deviations of static RTK positioning of the Huawei mate40 smartphone in the E, N, and U directions were 0.771 m, 0.910 m, and 0.925 m, respectively. The deviations of the Huawei mate40 smartphone for static RTK positioning in the E, N, and U directions were 0.498 m, 0.632 m, and 0.712 m, respectively. Thus, compared with conventional static RTK, robust static RTK improved the positioning accuracies by 35.4%, 30.5%, and 23.0% in the E, N, and U directions, respectively.

The positioning results of the low-cost M8 receiver obtained in the multi-system multi-frequency experiments are presented in Figure 6. The static RTK fixation rate was 97.4%, and it converged to within 1 m after 95 s. After robust processing, the convergence time was significantly reduced, and the fixation rate was increased to 99.8%. Robust static RTK positioning took only 13 s to converge to within 1 m, and the fixation difference was mainly reflected in the initial stage. Figure 6 shows the positioning result deviations of the low-cost M8 receiver in the multi-system multi-frequency experiments. The results of epochs 2800–5800 showed that the data quality before and after robust processing changed significantly. In particular, the positioning deviation in the E direction improved most obviously, and the positioning deviation reduced from 10 cm to 5 cm. The positioning accuracy in the other two directions also improved.

Table 6 shows the deviations of static RTK and robust static RTK positioning. The results showed that the deviations of the low-cost M8 receiver M8 in static RTK positioning in the E, N, and U directions were 0.104 m, 0.105 m, and 0.114 m, respectively. The positioning deviations after robust processing were 0.67 m, 0.072 m, and 0.075 m in the E, N, and U directions, showing increases of 35.5%, 31.4%, and 34.2%, respectively. The experimental results indicate that the positioning accuracy of the low-cost M8 receiver was better than that of the smartphones. In addition, the analysis of the static experimental results showed that robust RTK processing improved the positioning accuracy of the low-cost receiver more compared to the smartphones. Thus, it could be concluded that the improvement effect of robust static RTK processing in the positioning results with poor data quality was generally lower than that in the positioning results with higher data quality.

### 4.3. Analysis of Dynamic Robust Positioning Performance of Low-Cost Smart Terminals

This experiment was performed in a park forest area to analyze the robust RTK positioning performance in an occluded environment. In this experiment, the low-cost smart terminal and a professional receiver were fixed to each other, and the dynamic RTK experiment was performed by walking around the park three times so that the paths around the park could not completely overlap. The experimental time was 8:35–8:50 on 17 April 2023. Figure 7 shows the comparison charts of the motion trajectories of the Xiaomi 8, Huawei P40, Huawei mate40, and the low-cost M8 receiver. In the dynamic RTK graph in Figure 7, the yellow points represent the results of dynamic RTK positioning, and the blue points denote the results of robust dynamic RTK positioning. The results in Figure 7 show that the Xiaomi 8 smartphone had many offset points in the occluded environment, and the offset distance was large, with certain flying points. The positioning results of the Huawei P40 and Huawei mate40 smartphones were better than those of the Xiaomi 8 smartphone, but there were still offset points. In Figure 7, in the starting part of the lower left corner of the trajectory chart, it can be observed that the convergence performance of the Xiaomi 8 and Huawei P40 smartphones was weaker than that of the Huawei mate40 smartphone. The low-cost M8 receiver with the best data quality had stronger positioning results and a smaller overall path offset compared to the other devices.

After robust processing, the movement trajectory was adjusted, and the movement trajectory converged significantly, basically eliminating points with large positioning deviations. There were still certain offset points in the positioning results of the Xiaomi 8 smartphone, but there was no obvious offset in the other low-cost smart terminals after robust processing, and the points basically fitted the walking path. The positioning results of the low-cost M8 receiver did not have serious flying points, but the trajectory path had an offset. After the data were robustly processed, the offset trajectory of the path of all low-cost smart terminals was significantly corrected, especially the offset of all the path in the upper half of the image.

In the dynamic RTK positioning experiments, the low-cost intelligent terminal had poor elevation positioning accuracy due to the limitation of its own low-cost sensor, so this study mainly analyzed the plane accuracy. The walking dynamic RTK positioning error chart of the Xiaomi 8 smartphone is presented in Figure 8, where it can be seen that the positioning error in the E direction was generally within 1.7 m, and the positioning error in the N direction was generally maintained within 2 m. After robust processing, the positioning errors of the Xiaomi 8 smartphone were generally within 1.3 m and 1.5 m in the E and N directions, respectively. In Figure 8, dynamic RTK positioning is represented as scheme 1, and robust dynamic RTK positioning is represented as scheme 2.

Figure 9 shows the dynamic RTK positioning error diagram of the Huawei P40 smartphone. The results indicated that the positioning errors in the E and N directions were maintained within 1.5 m and 1.8 m, respectively. In the dynamic robust RTK positioning experiments, the positioning errors in the E and N directions were within 1 m and 1.5 m, respectively.

Further, Figure 10 shows the dynamic RTK positioning error diagram of the Huawei mate40 smartphone while walking. As shown in Figure 10a, compared to the previous two smartphones, the gross error of the dynamic RTK observation value was smaller, and the positioning deviations in the E and N directions were maintained at approximately 1.5 m and 1.7 m, respectively. The robust dynamic RTK positioning results were more stable than conventional RTK positioning. The positioning deviations in the E and N directions were within 1 m and 1.2 m, respectively.

The dynamic RTK positioning error diagram of the low-cost M8 receiver smartphone is shown in Figure 11a. The positioning error in both the E and N directions was within 0.4 m. Figure 11b shows the RTK positioning error after robust processing, and the positioning errors in the E and N directions were mostly maintained within 0.3 m and 0.35 m, respectively.

The positioning result statistics are given in Table 7, where it can be seen that the positioning results of the low-cost smart terminals increased by 33.1%, 33.2%, 42.5%, and 19.5% from top to bottom, and the convergence times increased by 41 s, 28 s, 40 s, and 17 s. Compared to the other devices, the low-cost M8 receiver could receive the largest number of satellite signals and provide better satellite spatial structure and redundant data. This result indicates that as the positioning accuracy of the low-cost smart terminals improved, the improvement effect of differential-resistant dynamic RTK positioning increased.

### 4.4. Experimental Performance Analysis of Low-Cost Smart Terminal In-Vehicle Dynamic Robust Positioning

An additional experiment was conducted to analyze the in-vehicle RTK positioning performance and examine the positioning performance of the smartphones. The experimental location was a closed highway section. The experiment time was 11:36–12:25 on 20 April 2023. In all figures presenting the experimental results, the yellow motion trajectory represents the RTK solution point, and the blue motion trajectory represents the robust RTK solution point. In Figure 12, Figure 13, Figure 14 and Figure 15, subfigures (a) and (b) represent the turning points and straight-line motion trajectories, respectively. Figure 12, Figure 13, Figure 14 and Figure 15 show the trajectories of the Xiaomi 8, Huawei P40, Huawei mate40, and low-cost M8 receiver RTK positioning, respectively. The positioning accuracy of the low-cost smart terminals was related to the selected path. Generally, the offset was larger when passing through areas blocked by tall objects.

As shown in Figure 12, the picture below presents the in-vehicle trajectory image of the Xiaomi 8. The positioning results of the Xiaomi 8 smartphone did not coincide with the actual route. The detailed enlarged view shows that some points had a large offset, regardless of whether it was a turn or a straight section. After robust RTK processing, the route convergence improved, the fitting degree with the actual route significantly improved, and the offset points were corrected.

The in-vehicle road map of the Huawei P40 smartphone is presented in Figure 13, where it can be observed that the original in-vehicle motion trajectory had better convergence than that of the Xiaomi 8 smartphone for both straight driving and turning. After robust RTK processing, the fitting degree with real data was significantly improved.

Figure 14 shows the in-vehicle route map of the Huawei mate40 smartphone. The original positioning results included data quality deviations in corners or heavily blocked in-vehicle routes. However, after robust RTK correction, the path fitting degree was improved.

Figure 15 shows the low-cost receiver’s vehicle road map, whose raw data had the highest positioning accuracy among all of the low-cost smart terminals. Regardless of whether the vehicle was turning or going straight, the positioning deviation was basically within the threshold range. Some positioning results at the corners had deviations that were too large, but after the correction by robust RTK positioning, they basically fitted the vehicle path.

The positioning accuracy results of the low-cost smart terminal for the in-vehicle RTK are presented in Table 8, where it can be seen that the accuracy of the Xiaomi 8 smartphone was improved by 18.7% after robust RTK positioning. Similarly, the accuracy of the Huawei P40 smartphone was improved by 29.1% after robust RTK positioning. In addition, the accuracy of the Huawei mate40 smartphone after robust RTK positioning was improved by 29.4%, and finally, the accuracy of the low-cost smart terminal M8 was improved by 13.8% after robust RTK positioning. It could be concluded that when the positioning result of the smart terminal was poor, the improvement effect of robust RTK positioning was more obvious as the positioning accuracy increased. However, when the original positioning accuracy was high, the improvement effect was significantly reduced.

## 5. Conclusions

This research adopted robust Kalman theory to perform robust RTK processing on the positioning results of low-cost smart terminals, and used measured data to analyze the positioning performance in static and dynamic modes. 

The following conclusions can be drawn based on the results obtained in this study:
(1)The results of the multi-system multi-frequency RTK positioning experiments indicate that the increase in the number of system frequencies can improve the positioning performance, and the increase in the number of equations in the multi-system multi-frequency positioning process plays an important role in ensuring the convergence and stability of the positioning results;(2)In the static RTK experiment with measured data, the robust RTK method improves the positioning accuracies of the Xiaomi 8 smartphone in the E, N, and U directions by 31.8%, 33.4%, and 32.8%, respectively; the positioning accuracies of the Huawei P40 smartphone in the E, N, and U directions by 31.4%, 31.5%, and 30.9%, respectively; the positioning accuracies of the Huawei mate40 smartphone in the E, N, and U directions by 35.4%, 30.5%, and 23.0%, respectively; and the positioning accuracies of the low-cost M8 receiver in the E, N, and U directions by 35.5%, 31.4%, and 34.2%, respectively;(3)The robust RTK positioning accuracy results of the low-cost smart terminals in the walking dynamic environment show that under walking conditions, the positioning accuracies of the Xiaomi 8, Huawei P40, and Huawei mate40 smartphones, and the low-cost M8 receiver improve by 33.1%, 33.2%, 42.5%, and 19.5%, respectively;(4)The positioning results of the in-vehicle experiment of the low-cost smart terminals show that the positioning accuracies of the Xiaomi 8, Huawei P40, and Huawei mate40 smartphones, and the low-cost M8 receiver devices increase by 18.7%, 29.1%, 29.4%, and 13.8%, respectively;(5)The analysis of the static and dynamic positioning results of the low-cost smart terminals shows that the improvements in the static RTK positioning results after robust processing is better than those of the dynamic RTK positioning results. Thus, the improvement effect on data results after robust processing is different for different positioning types. As the positioning accuracy of the low-cost smart terminals improves, the improvement effect of robust RTK positioning increases.

This study focused on low-cost, low-quality phase observation devices, revealed their performance limitations through in-depth experiments, and provides guidance for future improvements in the precision of positioning systems. The performance testing conducted in this study provides an initial understanding of how a low-cost device performs, but future research could delve deeper into the root causes of error accumulation and phase shift. Future research could extend the experimental objects to traditional real-time motion positioning terminals to comprehensively compare performance differences. In addition, positioning experiments of external antennas for smart terminals in complex environments could expand the understanding of the practical feasibility of this robust adaptive technology. Such comprehensive research is expected to deepen the understanding of precise positioning technology and advance the field.

## Figures and Tables

**Figure 1 sensors-24-01477-f001:**
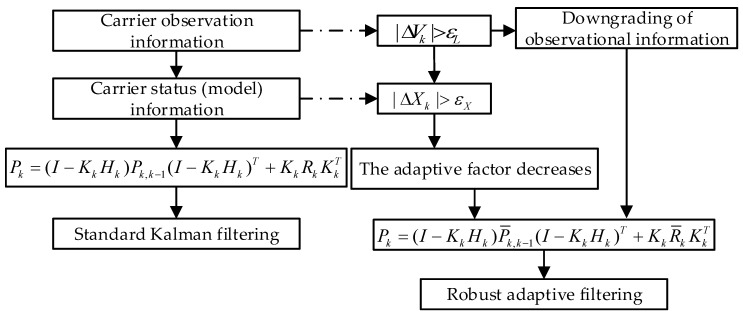
The flowchart of the robust adaptive Kalman filter.

**Figure 2 sensors-24-01477-f002:**
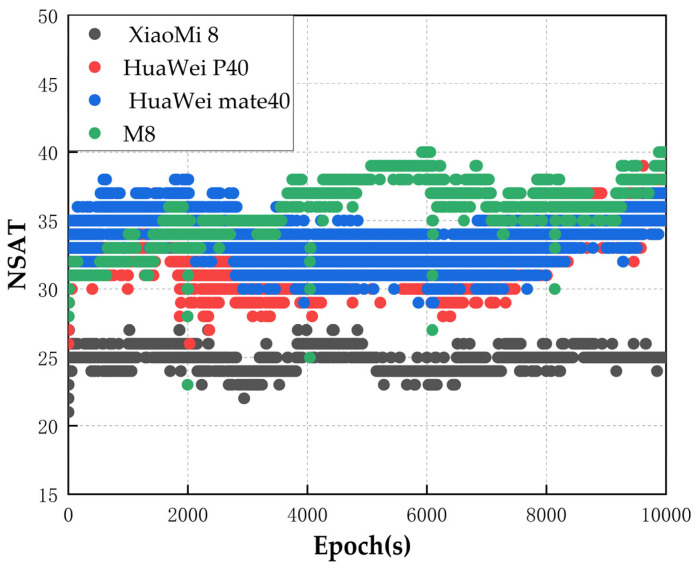
The number of satellites that could be detected by the low-cost smart terminals.

**Figure 3 sensors-24-01477-f003:**
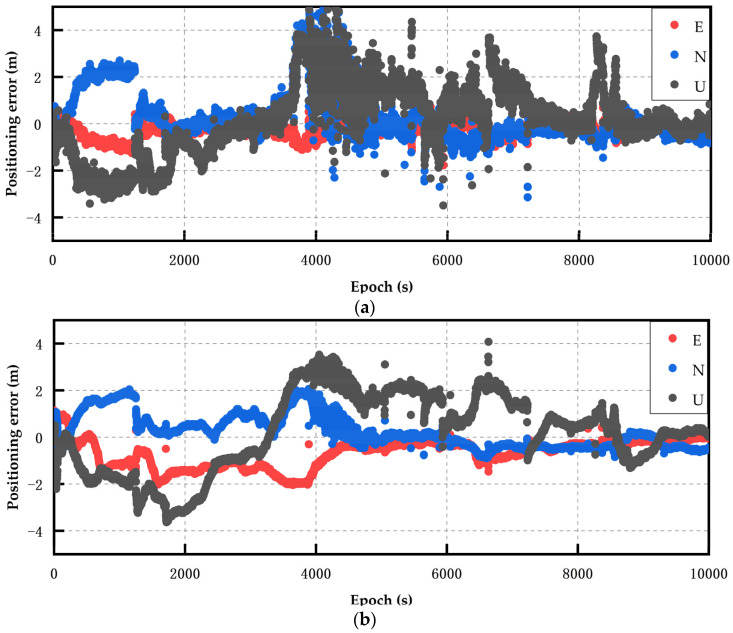
The Xiao Mi 8 smartphone’s static positioning error results. (**a**) Multi-system (GPS single-frequency + BDS single-frequency + Galileo single-frequency) RTK (scheme 1). (**b**) Multi-system (GPS single-frequency + BDS single-frequency + Galileo single-frequency) robust RTK (scheme 2).

**Figure 4 sensors-24-01477-f004:**
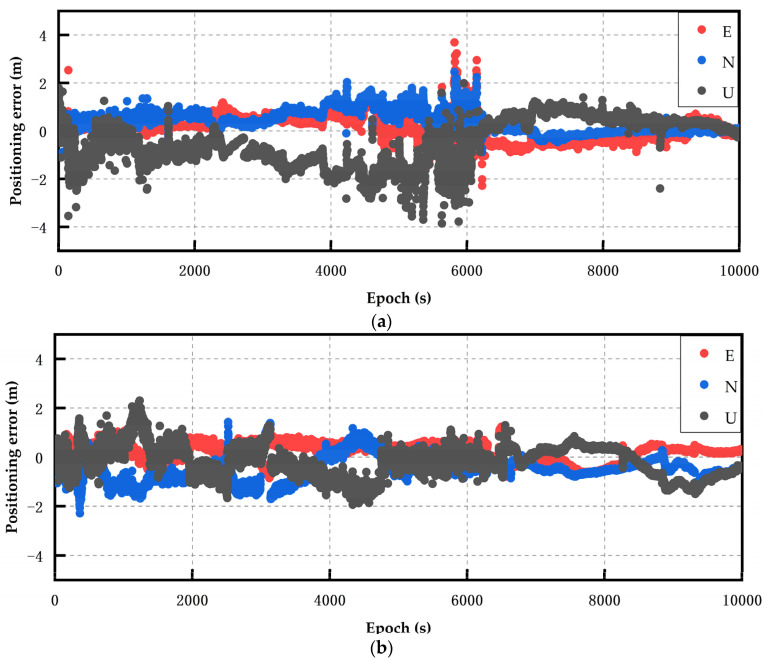
The Huawei P40 smartphone’s static positioning error results. (**a**) Multi-system (GPS double-frequency + BDS double-frequency + Galileo double-frequency) RTK (scheme 1). (**b**) Multi-system (GPS double-frequency + BDS double-frequency + Galileo double-frequency) robust RTK (scheme 2).

**Figure 5 sensors-24-01477-f005:**
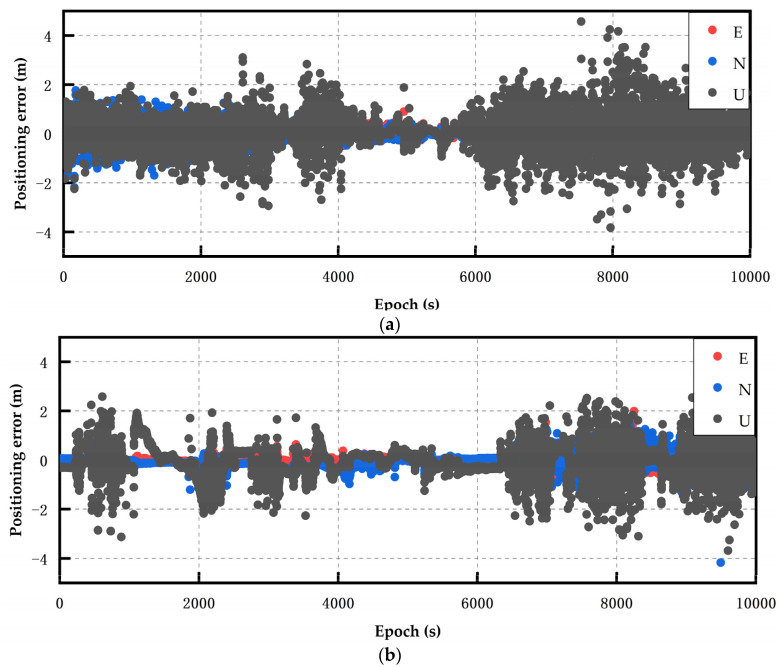
The Huawei Mate 40 smartphone’s static positioning error results. (**a**) Multi-system (GPS double-frequency + BDS double-frequency + Galileo double-frequency) RTK (scheme 1). (**b**) Multi-system (GPS double-frequency + BDS double-frequency + Galileo double-frequency) robust RTK (scheme 2).

**Figure 6 sensors-24-01477-f006:**
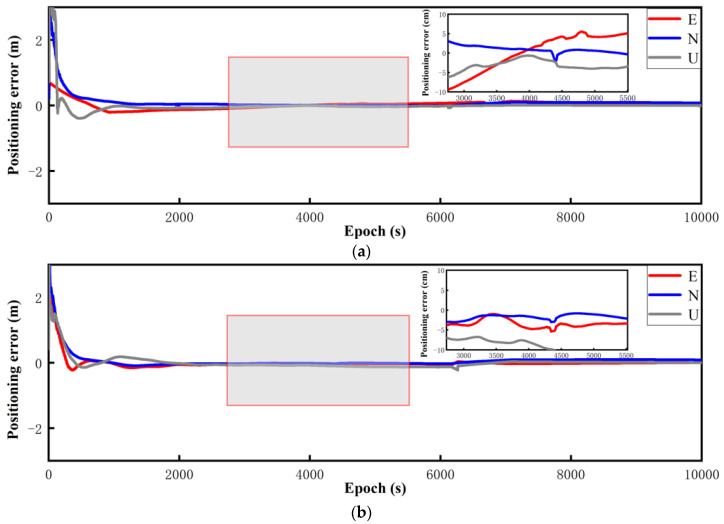
The positioning errors of the low-cost M8 receiver. (**a**) Multi-system (GPS double-frequency + BDS single-frequency + Galileo double-frequency) RTK (scheme 1). (**b**) Multi-system (GPS double-frequency + BDS single-frequency + Galileo double-frequency) robust RTK (scheme 2).

**Figure 7 sensors-24-01477-f007:**
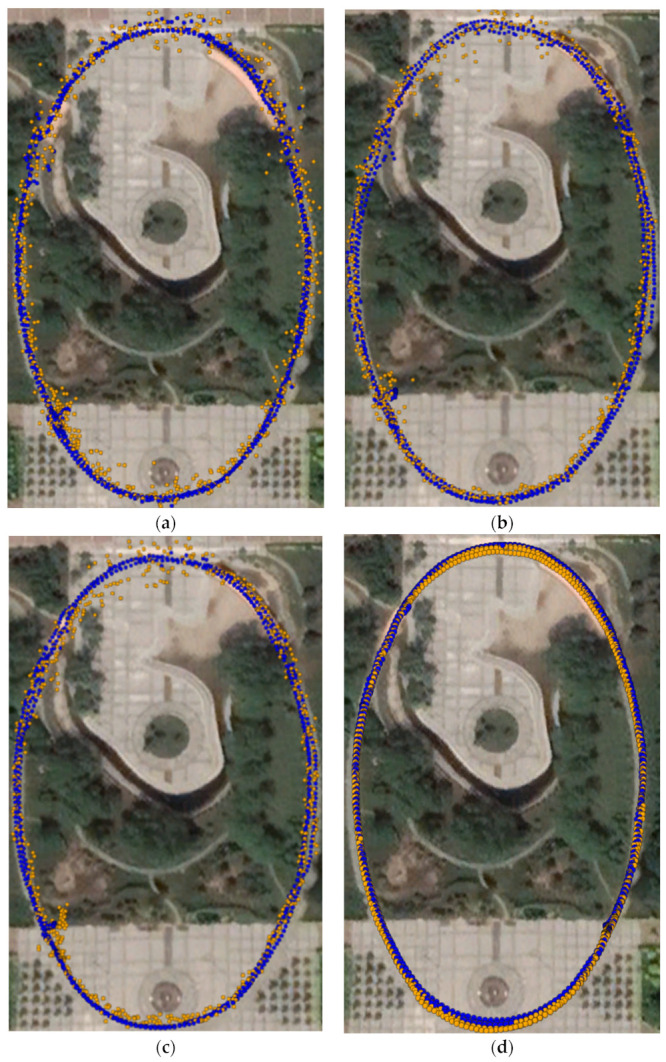
The low-cost smart terminal’s walking dynamic RTK positioning trajectories. (**a**) Xiaomi 8; (**b**) Huawei P40; (**c**) Huawei Mate40; (**d**) M8.

**Figure 8 sensors-24-01477-f008:**
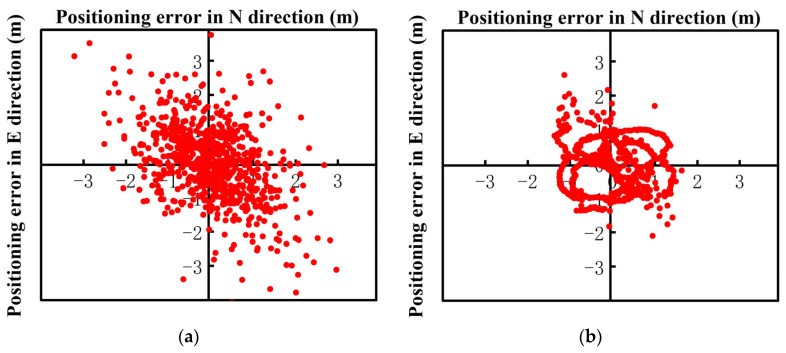
The walking dynamic RTK positioning errors of the Xiaomi 8 smartphone. (**a**) Xiaomi 8 Dynamic RTK (scheme 1); (**b**) Xiaomi 8 Robust Dynamic RTK (scheme 2).

**Figure 9 sensors-24-01477-f009:**
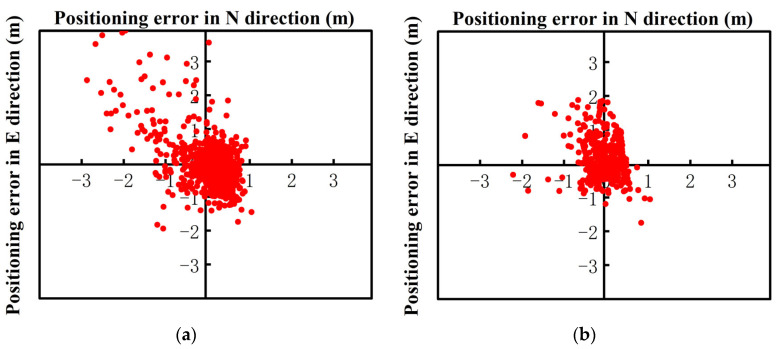
The walking dynamic RTK positioning errors of the Huawei P40 smartphone. (**a**) Huawei P40 Dynamic RTK (scheme 1); (**b**) Huawei P40 Robust Dynamic RTK (scheme 2).

**Figure 10 sensors-24-01477-f010:**
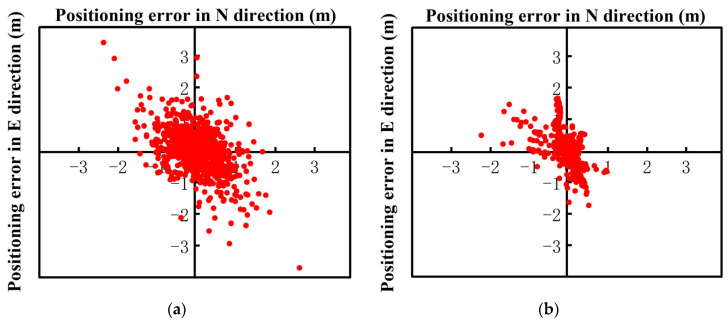
The walking dynamic RTK positioning error diagram of the Huawei Mate40 smartphone. (**a**) Huawei Mate40 Dynamic RTK (scheme 1); (**b**) Huawei 40 Robust Dynamic RTK (scheme 2).

**Figure 11 sensors-24-01477-f011:**
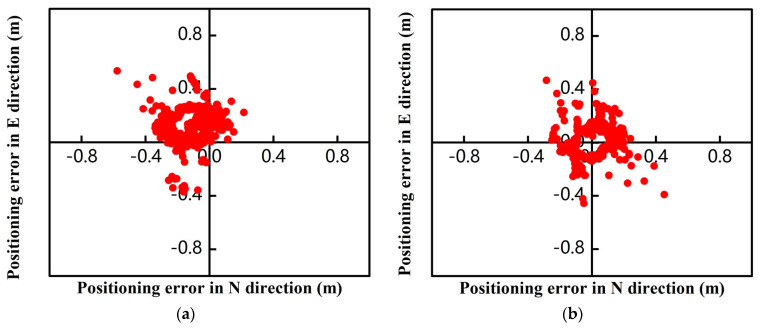
The walking dynamic RTK positioning error diagram of the M8 smartphone. (**a**) M8 Dynamic RTK (scheme 1); (**b**) M8 Robust Dynamic RTK (scheme 2).

**Figure 12 sensors-24-01477-f012:**
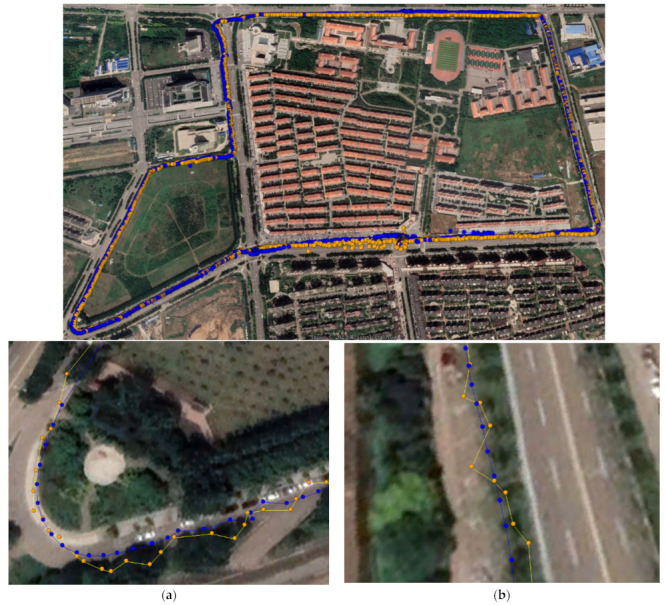
The Xiaomi 8 smartphone’s RTK positioning route. (**a**) At the turn; (**b**) Straight ahead.

**Figure 13 sensors-24-01477-f013:**
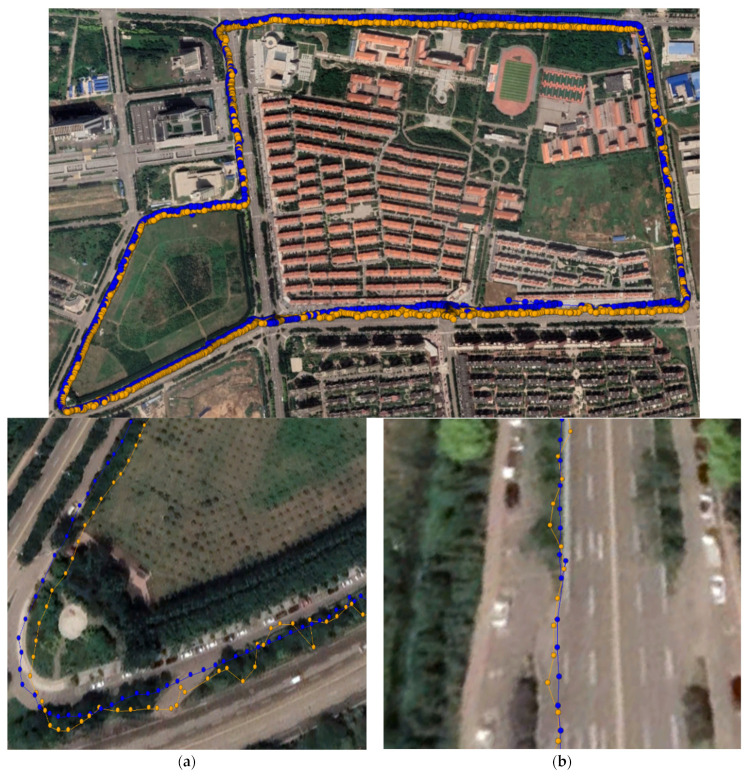
The Huawei P40 smartphone’s RTK positioning route. (**a**) At the turn; (**b**) Straight ahead.

**Figure 14 sensors-24-01477-f014:**
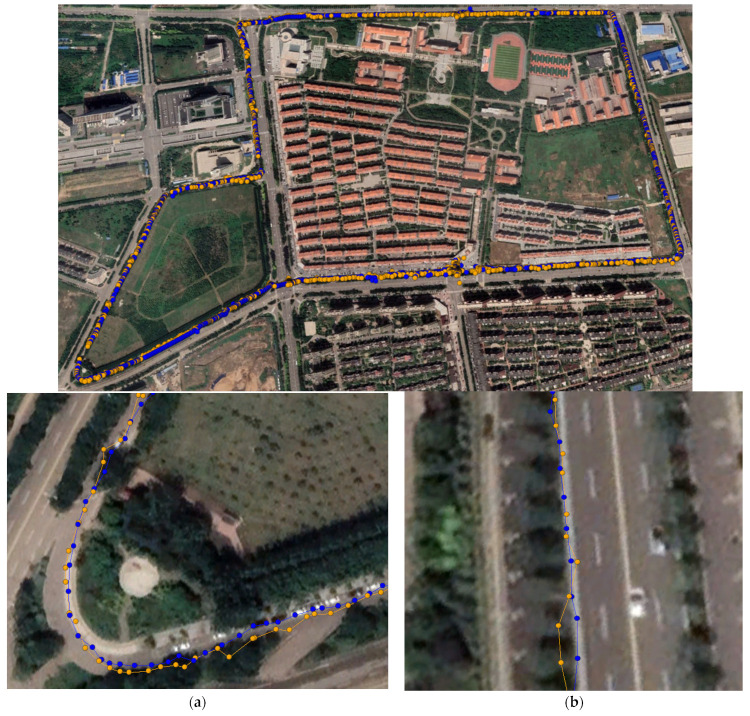
The Huawei Mate40 smartphone’s RTK positioning route. (**a**) At the turn; (**b**) Straight ahead.

**Figure 15 sensors-24-01477-f015:**
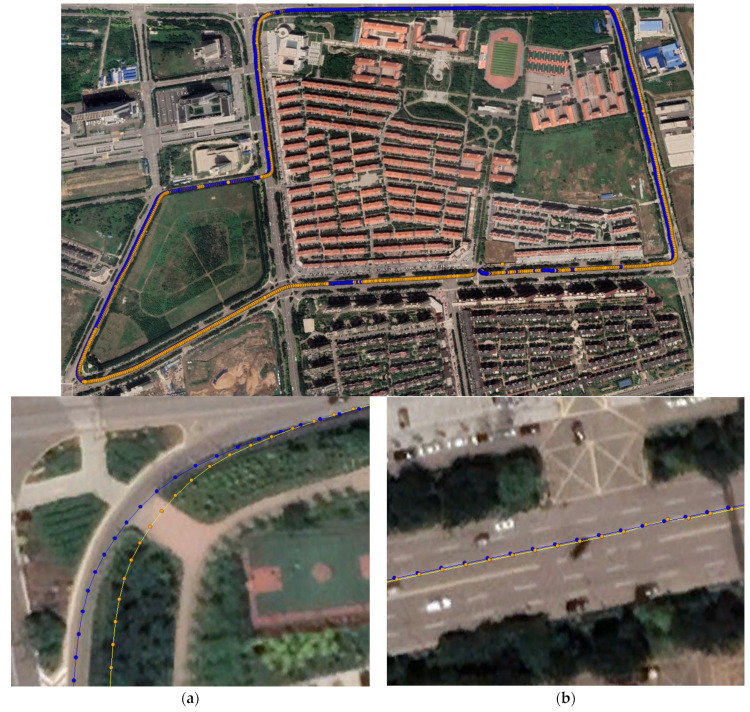
The low-cost smart terminal M8′s RTK positioning route. (**a**) At the turn; (**b**) Straight ahead.

**Table 1 sensors-24-01477-t001:** The device code, year of issuance, and chip types of the devices used in the experiments.

Type	Device Code	Year of Release	Chip Type
Xiao Mi 8	MI8B	2018	BCM47755
Huawei P40	P40B	2020	Kirin990
Huawei Mate40	Ocean	2020	Kirin 9000E
Low-cost receivers	M8	2020	Sinan K803

**Table 2 sensors-24-01477-t002:** The satellite systems and frequency bands that the devices used in the experiments could receive.

Type	GPS	BDS	Galileo	GLONASS	QZSS
Xiao Mi 8	L1/L5	B1I	E1/E5a	R1	J1/J5
Huawei P40	L1/L5	B1I/B1C/B2a	E1/E5a	R1	J1/J5
Huawei Mate40	L1/L5	B1I + B1C + B2a + B2b	E1 + E5a + E5b	R1	J1/J5
M8	L1/L5	B1I	E1	R1	J1

**Table 3 sensors-24-01477-t003:** The Xiaomi 8 smartphone’s static positioning result statistics (units: m). (RMS of E/N/U positioning errors and the convergence time).

Positioning Mode	E	N	U	Convergence Time (s)
RTK	0.86	0.79	1.07	126
Robust RTK	0.58	0.52	0.72	98

**Table 4 sensors-24-01477-t004:** The Huawei P40 smartphone’s static positioning result statistics (units: m). (RMS of E/N/U positioning errors and the convergence time).

Positioning Mode	E	N	U	Convergence Time (s)
RTK	1.10	1.01	1.53	112
Robust RTK	0.76	0.70	1.06	87

**Table 5 sensors-24-01477-t005:** The Huawei Mate 40 smartphone’s static positioning result statistics (units: m). (RMS of E/N/U positioning errors and the convergence time).

Positioning Mode	E	N	U	Convergence Time (s)
RTK	0.77	0.91	0.93	86
Robust RTK	0.49	0.63	0.71	63

**Table 6 sensors-24-01477-t006:** The low-cost M8 receiver’s static RTK positioning result statistics (units: m). (RMS of E/N/U positioning errors and the convergence time).

Positioning Mode	E	N	U	Convergence Time (s)
RTK	0.104	0.105	0.114	22
Robust RTK	0.067	0.072	0.075	13

**Table 7 sensors-24-01477-t007:** The smartphone walking RTK RMS results (units: m). (RMS of E/N/U positioning errors and the convergence time).

Type	RTK				Robust RTK			
	E	N	Plane	Convergence Time (s)	E	N	Plane	Convergence Time (s)
Xiao Mi 8	0.99	1.36	1.67	187	0.69	0.98	1.12	146
Huawei P40	0.91	1.05	1.39	162	0.56	0.75	0.93	134
Huawei Mate40	0.87	0.99	1.31	169	0.51	0.66	0.76	129
M8	0.13	0.15	0.16	61	0.11	0.11	0.13	44

**Table 8 sensors-24-01477-t008:** The smartphone riding RTK RMS results (units: m). (RMS of E/N/U positioning errors and the convergence time).

Type	RTK			Robust RTK		
	E	N	Plane	E	N	Plane
Xiao Mi 8	2.11	3.32	3.93	1.64	2.74	3.19
Huawei P40	2.08	2.12	2.98	1.46	1.52	2.10
Huawei Mate40	1.83	2.24	2.89	1.35	1.56	2.06
M8	0.72	0.91	1.16	0.63	0.87	1.07

## Data Availability

No new data were created or analyzed in this study. Data sharing is not applicable to this article.

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
