# Peer review of "Research on Robust Adaptive RTK Positioning of Low-Cost Smart Terminals"

_sensors, 2024, doi:10.3390/s24051477_

Round 1
Reviewer 1 Report
Comments and Suggestions for Authors
This manuscript was well organized with sufficient experiments and demonstration, and the results were reasonable and credible to support the proposed algorithm.
Suggestions were list bellow:
1. Abbreviations like 'IGGIII', 'OSR' should be explained at the first time.
2.The reference on line 61 should be [10], not [11]. And the reference [1] and [23] were different with other references. Please also checked the citation and expression of all the other references.
3.There is not need to keep 3 numbers after decimal point since the deviations were around meters level.
4. On line 471, the 'errors' should be 'results', please check if there were other typo errors
5. The Figure 2-5 could be improved by plot with 'grid on' X and Y axis, also the points collection line can be cancelled.
Comments on the Quality of English Language
The quality of the English is acceptable except for some 'Chinglish' expressions.
Author Response
Thank you for your letter and for the reviewers’ comments concerning our original manuscript entitled “Research on Robust Adaptive RTK Positioning of Low-cost Smart Terminals” (ID: sensors-2859178). Those comments are all valuable and very helpful for revising and improving our paper, as well as the important guiding significance to our researches. We have studied comments carefully and have made correction which we hope meet with approval. Revised portion are marked the “redline” comparison feature in the paper.

Reviewer 2 Report
Comments and Suggestions for Authors
The proposed manuscript deals with actual problems, necessary for robotic systems and Internet of transport. Here the choice of robust Kalman filtering seems to be adequate for noisy signals procession in GNSS tasks, and its potential is far from being exhausted now. RTK methods has also shown themselves to be useful data procession technology for navigation tasks.
Novelty of the paper refers mainly to multi-system multi-frequency RTK positioning, where multi-equation modelling with the help of robust Kalman filtering provides improvement of convergence and stability. Interesting results refer to experiments held for various smartphones types and their comparison with low-cost smart terminals. The last one platform has shown behaviour somewhat different from smartphones.
Supporting literature review contains adequate citations and corresponds to research topic.
Data obtained analysis refer both to walking and in-vehicle testing. Used model has provided substantial improvement of positioning accuracy, and these results supports possible further applications of robust RTK scheme.
Conclusions correspond to given materials.
The text is clear enough, but some corrections are possible.
Some drawback of the paper is that long enough description of the method seems to be not very clear and needs some corrections, as the authors didn`t give briefly in the introduction part the general scheme of data procession and the sequence of main steps, what would be useful for the complicated method. But such brief scheme would help to understand quickly, where basic methods of RTK and robust Kalman filtering were used, and where is the gain of innovations introduced by the authors. Also, used equations for RTK and robust Kalman filtering were not supported by detailed citations, but given commentaries refer to selected specific peculiarities.
Recommended corrections.
11. The very basic idea of RTK method and robust Kalman filtering is to be briefly commented just in sec.1, as the novelty introduced by authors can be formulated, e.g. in sec.2 (including static and robust versions of RTK). That will help to understand quickly the sec.3.2 without repeated browsing of the paper.
22. Please discuss or comment somehow the possible influence of additional external noise level on the results of the proposed method, as readers may have quite traditional questions concerning colored noise for Kalman method.
33. It is desirable to comment somehow the used software or necessary products for tested mobile platforms.
44. Please, explain why elevation positioning accuracy of low-cost smart terminals was poor for quite an advanced method (row 551).
55. Disclose the abbreviation OSR and give some clear commentary to it after the row 160. Also the commentary to eq.(6) is not very clear, and explain here, please, why its ambiguity parameter is integer.
66. The explanation to eq.(10) (rows 184-194) is not very clear.
77. Three satellite systems G, C,R were used in eq. (16), but If this number can be enlarged ?
88. Comment briefly the difference between the Huber function and the chosen IGGIII for rows 244-246.
99. Refine the criteria for the convergence time in Tables 3-6.
110. Disclose abbreviation BDS-3 after the first use (row 202) and give some citation. Also, data in Table 2 concerning different positioning systems need citations and precision parameters.
111. Literature citations would be useful for eq.(30) and for the sec. 4.1, concerning Doppler speed measurements and LAMBDA method.
Comments on the Quality of English Language
Moderate corrections are possible.
Author Response

(The authors gave the same response as above.)

Reviewer 3 Report
Comments and Suggestions for Authors
The authors use RTK and KF techniques for high-performance localization in smartphones, and the proposed algorithms are mathematically feasible, but the paper still needs to explain some important issues.
Major
1. RTK is a local positioning method where the source of the reference information must be near the user. How can the reference information be transmitted to the smartphone in practical application? Firstly, the communication base station itself does not have this function. Secondly, although the reference information can be transmitted through the data channel of BS within the coverage area of cellular network, there is no cellular network coverage when operating in the field. At this time, if you want to use BT or WiFi, the communication distance is very small, and if you set up cellular BS by yourself, the communication distance will not be very large, and it is illegal to set up BS privately.
2. What is the positioning correction algorithm used by navigation software on cell phones nowadays, and what are the advantages of the author's positioning accuracy compared to that of commercial navigation software?
3. In the introduction section, authors are required to clearly state what their contribution is compared with previous research.
Minor
1. RTK is usually implemented with dedicated terminals, and the authors propose that cell phones implement RTK, is this necessary?
Author Response

(The authors gave the same response as above.)
